



# Novel oxalate-carbonate pathways identified in the tropical dry evergreen forest of Tamil Nadu, India

Camille Rieder[1,2], Eric P. Verrecchia[1], Saskia Bindschedler[2], Guillaume Cailleau[2], Aviram Rozin[3], Munisamy Anbarashan[4], Shubhendu Dasgupta[3], Thomas Junier[5], Nicolas Roeschli[1], Pascal Vittoz[1], Mike C. Rowley[6]

[1]Institute of Earth Surface Dynamics, University of Lausanne, Lausanne, 1015, Switzerland
[2]Microbiology Laboratory, University of Neuchâtel, Neuchâtel, 2000, Switzerland
[3]Sadhana Forest, Auroville, 605101, Tamil Nadu, India
[4]Ecology Department, French Institute of Pondicherry, Pondicherry, 605001, India
[5]Swiss Institute of Bioinformatics, Swiss Federal Technology Institute of Lausanne, Vital-IT Group, Lausanne, 1015, Switzerland
[6]Department of Geography, University of Zurich, Zurich, 8057, Switzerland

*Correspondence to*: Camille Rieder (Camille.Rieder@unil.ch)

**Abstract.** The tropical dry evergreen forest (TDEF) is a vital but endangered ecosystem in India, crucial for supporting cultural services, biodiversity, and organic carbon storage. The oxalate-carbonate pathway (OCP) is an understudied process in which plants and oxalotrophic microorganisms convert atmospheric $CO_2$ into calcium carbonate ($CaCO_3$) within plant tissues or tree-adjacent soils. Yet, despite its significance, the OCP has not been studied in the TDEF of India. This study aimed to assess novel OCP systems associated with three TDEF diagnostic species (*Diospyros ebenum, Lepisanthes tetraphylla*, *Sapindus emarginatus*) and one local agroforestry species (*Artocarpus heterophyllus*) in the restored- and primary-TDEF of Tamil Nadu. Surface soil samples (0-10 cm) were collected from an adjacent and control distance away from trees, along with tree biomass samples, and investigated for oxalate production (microscopy and enzymatic assays), oxalotrophic microbial communities (*frc* gene sequencing), and tree-induced shifts in soil biogeochemistry. Oxalate was detected in all species (4.4±3.2 % dry weight), accompanied by $CaCO_3$ precipitation on biomass. Oxalotrophic microbial communities were dominated by Actinomycetota (86 %), which were also identified in electron micrographs. Soil biogeochemical shifts indicative of active OCPs were also observed, particularly in the hollowed-out trunks of the TDEF trees. However, differences between adjacent and control soils were less pronounced, suggesting that monsoon conditions leached OCP precipitated $CaCO_3$ from the adjacent soils. This research provides the first evidence of active OCPs in Indian TDEF, highlighting a previously unrecognized mechanism for organic and inorganic carbon cycling in this threatened ecosystem.



# 1 Introduction

Tropical dry evergreen forests (TDEF) are ecosystems of significant importance but are also endangered. The only remaining intact TDEF is located in mainland Southeast Asia (Wikramanayake et al., 2002), while smaller fragmented forests can be found in the Neotropics, some regions of Africa, Sri Lanka, and the Coromandel coast of India (Parthasarathy et al., 2008; Sudhakar Reddy et al., 2018; Udayakumar and Parthasarathy, 2010). Although the exact definition of TDEF is subject to debate (Everard et al., 2018), the term is hereafter retained to describe the TDEF of the Coromandel coast due to its widespread use in the literature, although the ecoregion is also referred to as the East Deccan dry-evergreen forest (Wikramanayake et al., 2002). The TDEF is an endemic ecosystem dominated by drought-resistant species, which form an evergreen canopy with a low maximum height (9-12 m; Venkateswaran and Parthasarathy 2003; Parthasarathy et al. 2015). It is considered the rarest type of forest ecosystem on the Indian subcontinent (Blanchflower, 2005) with estimates suggesting that only ~4 % of its original extent are remaining (Wikramanayake et al., 2002). Yet, of these forest remnants, 95 % are described as degraded shrublands and only 5 % (~0.2 % of the original cover) as pristine or primary forests (typically preserved within 'Sacred Grove' areas; Kent 2013). The remaining portion of the TDEF is subject to a number of anthropogenic pressures (Sudhakar Reddy et al., 2018), including the expansion of settlements, increased rubbish-tipping, grazing, fires, logging, and the introduction of invasive species (Narasimhan et al., 2009; Parthasarathy and Karthikeyan, 1997; Venkateswaran and Parthasarathy, 2003), which further transform the forest into scrublands (Anbarashan and Parthasarathy, 2013; Champion and Seth, 1968; Puri et al., 1989). Consequently, India's TDEF is considered a key ecosystem, though it has suffered extensive degradation and continues to face serious threats from human activity.

The TDEF is widely recognized for its role in supporting vital ecosystem services. Previous studies have emphasised its significance for cultural services (Kent, 2013; Ramanujam and Praveen Kumar Cyril, 2003), the production of food, fuel, fibre, and medicinal products (Everard et al., 2017), providing habitat for fauna (David et al., 2015; Frignoca et al., 2021) and plant biodiversity (Anbarashan et al., 2020; Anbarashan and Parthasarathy, 2013), storm buffering (Everard et al., 2017), climate regulation (Everard et al., 2018), as well as for organic carbon (C) storage (Pandi and Parthasarathy, 2015; Yadav et al., 2022). Despite its relatively low canopy height, the TDEF is a dense C reservoir, with stock estimates in aboveground biomass alone ranging between 39.7-576.4 t C ha$^{-1}$ (Mani and Parthasarathy, 2007; Pandi and Parthasarathy, 2015). Comparatively, recent estimates for agroforestry landscapes in Southern India, with taller canopies (in Kerala: 34.8-558.5 t C Ha$^{-1}$; George et al. 2025), suggest that the TDEF has a similar C storage capacity. Everard et al. (2018) estimated that reforestation of degraded shrubland to dense TDEF has the potential to sequester significant amounts of organic C, estimated at 292 Mg C ha$^{-1}$, when accounting for the C storage in soil organic carbon stocks (120 Mg C ha$^{-1}$) and aboveground biomass (172 Mg C ha$^{-1}$). Thus, it is crucial to enhance our understanding of this endangered natural resource to safeguard its ecological functions and biodiversity, and to better establish the potential of TDEF reforestation for C storage.

The oxalate-carbonate pathway (OCP) is an interconnected plant-soil-microbe biogeochemical system, which can sequester significant quantities of atmospheric $CO_2$ as inorganic C, yet it remains unexplored in the context of the TDEF. The OCP





begins when specific plant or fungal species produce oxalic acid ($pK_{a1}$ - 1.3, $pK_{a2}$ - 4.3), which can then be converted into calcium oxalate crystals (CaOx; $CaC_2O_4.nH_2O$). CaOx is considered the second most abundant biomineral (Krieger et al.,

2017). In plants, CaOx crystals are produced in specialised cells termed idioblasts (Frey-Wyssling, 1925, 1981; Nakata, 2002, 2003) while in fungi they are usually observed coating the outer part of hyphae (Arnott, 1995). The primary function of these crystals in plants is thought to be the regulation of cytoplasmic calcium concentrations or protection against herbivory, but multiple secondary phyto-functions have also been proposed (Franceschi and Nakata, 2005). In fungi, oxalic acid is a multifunctional metabolite that can regulate the bioavailability of mineral elements, function as a virulence factor in

plant-pathogenic fungi, and serve as a chelating agent in lignocellulose degradation processes (Dutton and Evans, 1996). Yet, despite the widespread production and low solubility of CaOx (monohydrate solubility constant between 1.77 and $6.7 \times 10^{-9}$ at 25°C in ultrapure water; Ibis et al. 2020), the crystals are rarely measured in high concentrations (> mg kg$^{-1}$) in soils adjacent to large oxalogenic trees (Álvarez-Rivera et al., 2021; Certini et al., 2000).

Instead of accumulating in soils, when plant-associated CaOx crystals are released during litter decomposition (Hervé et al.,

2018) or belowground *via* root turnover or exudation (Cailleau et al., 2014; Rowley et al., 2017), they can readily be metabolised by oxalotrophic microorganisms. Oxalotrophs have been described in α-, β-, and γ-Proteobacteria (gram-negative), Firmicutes, and Actinomycetota (gram-positive; Sahin 2003), which metabolise oxalate either obligately (as a required C and energy source) or facultatively (as an alternative C and energy source). C assimilation during oxalotrophy occurs through the glyoxylate cycle or the serine pathway (Bassalik, 1913; Palmieri et al., 2019; Sahin, 2003). The

metabolism of oxalate induces a distinct localised alkalinisation of the soil by converting a stronger acid ($H_2C_2O_4$) into a weaker one ($H_2CO_3$). If the pH increase exceeds the stability pH of calcium carbonate ($CaCO_3$), it can enable its precipitation from the byproducts of oxalotrophy, adjacent to an CaOx-producing tree species (Braissant et al., 2002; Verrecchia et al., 2006). If this sequestration involves Ca that is not derived/liberated from primary $CaCO_3$, then the process can be considered a net C sequestration (Cailleau et al., 2014; Rowley et al., 2017). Consequently, interactions between CaOx-producing tree

species and oxalotrophs can actively drive an OCP, leading to the sequestration of atmospheric $CO_2$ as inorganic C.

The OCP was first reported in association with *Milicia excelsa* (Welw.) C.C.Berg (family Moraceae; Carozzi 1967). Direct measurements of the *M. excelsa* system suggested that an individual tree can sequester around 979 kg $CaCO_3$ over its lifetime (Braissant et al., 2004; Cailleau et al., 2004, 2005, 2011). These field observations have been supported by 1-dimension (Gatz-Miller et al., 2022) and 2-dimension (Gatz-Miller et al., 2023) reactive-transport models, which suggest that

$CaCO_3$ precipitation in an acidic, active *M. excelsa* OCP system begins within ~20 years and continues throughout the tree's lifespan. Active OCPs have been documented worldwide (see Supplementary S.Table 1), in South (Cailleau et al., 2014), in the Caribbean and Central America (Álvarez-Rivera et al., 2021; Rowley et al., 2017), and North America (Garvie, 2006, 2003), numerous African countries (Aragno and Verrecchia, 2012; Cailleau et al., 2005; Hervé et al., 2021; Pons et al., 2018), in Israel (Verrecchia, 1990), and in India (Hervé et al., 2018). Hervé et al. (2018) investigated the OCP associated

with *Terminalia bellirica* (Gaertn.) Roxb. (family Combretaceae) in Central India (Khajuraho, Madhya Pradesh), demonstrating a significant $CaCO_3$ accumulation in the species' bark (82 % $CaCO_3$ dry weight [D.W.]). Yet, despite the high





concentrations of CaCO$_3$ in the bark of *T. bellirica*, there were low increases in the CaCO$_3$ content of tree-adjacent soils (15 g kg$^{-1}$), relative to control soils (5 g kg$^{-1}$ CaCO$_3$), emphasising the need for more research to improve understanding of the OCP in India. Overall, CaOx is an abundant biomineral (Krieger et al., 2017) and oxalotrophy is a widely distributed metabolism, with functional oxalotrophic communities also identified worldwide (Bravo et al., 2013; Cowan et al., 2024; Hervé et al., 2016), thereby highlighting the potential global distribution of OCP ecosystems and their potential presence in the TDEF. We thus hypothesise that the OCP may represent a substantial but underexplored C sink in India's TDEF ecosystems, which has so far remained undetected.

To explore potential OCP ecosystems within restored- and primary-TDEF in Tamil Nadu, India, an initial field survey was conducted. This led to the selection of four tree species: three typical of the TDEF, *Lepisanthes tetraphylla* (Vahl) Radlk. (family Sapindaceae), *Sapindus emarginatus* Vahl (family Sapindaceae), *Diospyros ebenum* J.Koenig (family Ebenaceae), and one widely used in regional agroforestry practices, *Artocarpus heterophyllus* Lam. (family Moraceae). Optical and scanning electron-microscopy were then used to image CaOx within plant biomass and CaOx contents were quantified using spectrophotometric enzymatic oxalate kit analyses. The composition of the oxalotrophic microbial community in association with TDEF diagnostic species was assessed in the roots, litter, and nearby soils using high-throughput sequencing (HTS) of the *frc* gene. Tree-induced shifts in the soil biogeochemistry were measured by sampling surficial (0-10 cm) tree-adjacent (at the bottom of the tree), and control soils (20 m of distance), and then quantifying their soil pH, exchangeable cations, organic C, total N, and CaCO$_3$ equivalent content. Through these analyses, we identified four novel OCP ecosystems in the TDEF; however, despite significant CaCO$_3$ deposits on the tree biomass, differences in biogeochemical properties between the adjacent and control soils were less pronounced.

## 2 Material and methods

Hereafter supplementary materials are denoted with a S (S.Fig, S.Table, S.Methods, and S.Eq).

### 2.1 Site setting

The study area is located north of Puducherry city (formerly Pondicherry), around Auroville, on the Coromandel Coast in southeastern India (Fig. 1A). The climate of this region is classified as Tropical Savannah with a dry winter (Aw in the Köppen-Geiger classification), with an average temperature of ~28.6°C, ranging between 24.8-30.8°C (January and May, respectively). The region receives an average rainfall of ~1225 mm yr$^{-1}$, with the majority occurring during the northeast monsoon between October and December, and a smaller contribution for the southwest monsoon between July and August (Duraisamy Rajasekaran et al., 2024).



## 2.2 Field campaign

### 2.2.1 Field screening

A field campaign was conducted in the study region between September and December 2019. An initial screening of tree species in the field to identify active OCP ecosystems was completed in accordance with the methods outlined by Cailleau et al. (2014). Briefly, for each potential tree species, the presence of above- and below-ground $CaCO_3$ was detected with 10 % hydrochloric acid (HCl). Differences in soil pH between adjacent- and control-surface soils, sampled 20 m away from the tree, were assessed using pH Hellige tests. CaOx in leaf tissues was detected in the field using light microscopy, by staining it in plant tissues with Carnoy fluid (Ilarslan et al., 2001). From this initial investigation, three species typical of the TDEF (*D. ebenum*, *L. tetraphylla*, and *S. emarginatus*), together with *A. heterophyllus*, a Southern Indian species from the Western Ghats widely used in local agroforestry systems were selected. Four individuals of each of these species, two smaller and two larger, were then sampled. Each individual will hereafter be labelled with the first letter of the species name and an increasing number from 1-4, with the two smaller individuals being labelled 1-2, and the two larger individuals 3-4 (e.g., *D. ebenum* = D1 to D4). Details of the specific size of individuals is listed in S.Table 2.

The study investigated trees of a primary TDEF at Puthupet Sacred Grove, Keezhputhupattu (Fig. 1C), a TDEF reforestation site at Sadhana Forest in Auroville (Fig. 1D), and a few smaller sampling locations around Auroville such as Aurodam (Fig. 1B; individual S1), Pitchandikulam Forest (Fig. 1B; individual S2), Kuilapalayam (Fig. 1B; individual S4), and Lake Estate Farm (Fig. 1B; individuals A3 & A4). Tree heights were quantified in the field using a clinometer and tape measurements, while their diameters at breast height (DBH) were calculated from trees' circumferences at 1.3 m from the soil surface. As there is no universal allometric equation for TDEF species, calculations were performed using available species-specific data (Anil and Parthasarathy, 2016; Brown et al., 1989; FSI, 1996; Mokany et al., 2006; Murphy and Lugo, 1986).





**Figure 1. Map of the study area, tree locations, and underlying parent material adapted from a base map outline (© Google Maps, 2023; d'Ozouville et al., 2006). (A) Location of Tamil Nadu, within India. (B) Tree locations around Auroville, north of Puducherry, India; see tree and site details in S.Table 2. (C) The Puthupet Sacred Grove sampling site. (D) The Sadhana Forest sampling site. (E) Stratigraphic cross section of local geological units showing the eastward tilt of the underlying strata and locations of sampling sites C & D, relative to the strata, adapted from (d'Ozouville et al., 2006).**

The parent material of the study sites comprised three principal sedimentary formations, the Alluvium deposits, Cuddalore sandstone, and Manaveli clay formations (Fig. 1E; d'Ozouville et al. 2006), for which, details can be found in the S.Methods section. At the sampling sites, various Reference Soil Groups (IUSS Working Group WRB, 2015) were described, primarily influenced by their respective parent materials. Arenosols were identified in the Alluvium deposits, Ferralsols or Leptosols in the Cuddalore sandstone, Plinthosols at the transition between the Cuddalore sandstone and Manaveli clay, and Regosols or Calcisols in the underlying impermeable Manaveli clay.





### 2.2.2 Sampling and preparation

In November 2019, a total of 83 samples of plant tissues and 35 soil samples were collected from four individuals of each tree species and 45 additional samples for microbiological analyses ($n$=163). The samples included bark, branches, leaves,
litter, roots, adjacent and control surficial soil samples from each individual. We were limited in our ability to dig entire soil profiles around the trees because the sites were typically preserved in the traditionally revered 'sacred groves'. Hence, with explicit permission, surficial samples were instead collected to a depth of 10 cm near the trees (adjacent soil) and at a control distance of 20 m away from the tree (control soil). Additionally, where feasible, the upper 10 cm of soil from hollowed-out trunks of three individuals (D3 - *D. ebenum*, L4 - *L. tetraphylla* and S3 - *S. emarginatus*) were also sampled. All plant and
soil samples were dried at 40°C for a minimum of 24 h.

For assessing the oxalotrophic guild via HTS, separate 3 g samples of litter, roots, rhizospheric soil (soil directly surrounding the root interface), adjacent soil, and control soil were collected in triplicate from one individual of each TDEF species (D3 - *D. ebenum*, L4 - *L. tetraphylla*, S3 - *S. emarginatus*; $n$=45). Samples were collected with a sterilised spatula and preserved for transport and sequencing by immediately submerging them in 5 mL Qiagen Biostability RNAlater. All samples were sent
to the University of Lausanne (Switzerland) and samples for HTS were then transported to the University of Neuchâtel (Switzerland).

Soil samples were sieved to 2 mm at the University of Lausanne. A subsample of each sample was ground. Soil subsamples were processed in a FRITSCH Pulverisette 7 planetary mill at 600 rpm for 3 min, while dry plant subsamples were ground in a FRITSCH Pulverisette 14 rotor mill at 10,000 rpm for 3 min.

### 2.3 Characterization and quantification of biominerals

The characterization of CaOx crystals was conducted through microscopy observations, while the quantification was performed using enzymatic oxalate kit analyses. Biominerals were imaged and characterized using a scanning electron microscope coupled with an energy-dispersive X-ray spectrometer (SEM-EDX; Tescan Mira LMU and Penta-FET 3x detector). Ground plant material was loaded onto alloy stubs using adhesive stickers and AuPd sputter coated (Cressington
108 auto) for 1 min. Images were captured at 5 or 10 kV acceleration voltage with an average working distance of 12 mm. Quantitative EDX analyses were performed at 20 kV, and peaks were analyzed using the Oxford Instruments Aztec software.

The CaOx content of soil or biomass were quantified colourimetrically using oxalate enzymatic assay kits (LIBIOS). CaOx was extracted according to Certini et al. (2000). Briefly, 0.1 g of ground plant material or 2 g of ground soil was combined
with 5 ml 1 *M* HCl and shaken on a rotary table for 16 h at 150 rpm. The extractions were then centrifuged (1560 g for 5 min), separating the supernatant, before diluting (1:5), and neutralising the extracts (pH = 5-7). After this, the procedure followed the kits instructions, and the CaOx content was measured on a PerkinElmer UV/VIS Lambda 365 spectrophotometer at 590 nm. Measurements were made relative to anhydrous oxalic acid standards, which were measured at



the beginning and end of each run. In addition, analytical replicates and blanks were measured to check for analytical drift,
reproducibility, and background contamination, respectively. A molar correction factor was applied based on the molecular
weight of CaOx monohydrate, which was the most observed crystal habit in the microscopy (142.11 g mol$^{-1}$), to convert
extracted values to CaOx % oven dry weight (D.W.) of plant material.

## 2.4 Soil analyses

Soil analyses results were corrected according to their hygroscopic moisture content. Concentrations or contents are thus
reported based on oven dried soil weight (105°C; van Reeuwijk 2002). Blind replicates were measured in all sample analyses
(>20 %) to ensure the reliability of the results. See the S.Methods section for the reasoning regarding each specific method.
Soil pH$_{KCl}$ was measured potentiometrically with a glass-body combination electrode (Thermo Scientific™ Orion Star AIII)
at a 1:2.5 ratio in 1 $M$ KCl (Pansu and Gautheyrou, 2006). The extraction of exchangeable cations (Ca$^{2+}$, Mg$^{2+}$, Na$^+$, K$^+$, and
Al$^{3+}$) was conducted using 0.0166 $M$ cobalt hexamine extraction (Aran et al., 2008). A 2 g portion of the sample was mixed
with 40 mL of 0.0166 $M$ cobalt hexamine solution and shaken on a rotary table at 120 rpm for 1 h. The mixture was then
filtered (Whatman No. 42), diluted (1:25) in 2 % HNO$_3$, and measured on a Perkin Elmer Optima 8300 inductively coupled
plasma-optical emission spectrometer with an internal Sc standard to correct data for analytical drift. The total element
composition of ground and furnaced (Solo 111-13/10/30) samples, fused into lithium tetraborate pellets (PANalytical Perl
X3 Fuser), was quantified using X-ray fluorescence (PANalytical Axiosm spectrometer). The spectrometer was equipped
with a 4 kW Rh X-ray tube and calibrated with 21 international silicate rock reference materials. The results were corrected
for loss of ignition. The bulk mineral composition of soil samples was analyzed using X-ray diffraction (X-TRA Thermo-
ARL Diffractometer) with the methods described in Adatte et al. (1996). The powder holders were prepared with pressed
ground soil, and diffractograms were transformed using the Thermo Fischer WinXRD 2.0-6 program. The mineral peaks
were then converted into percentages according to Adatte et al. (1996).
The total CaCO$_3$ equivalent content was determined using the acetic acid dissolution technique (Loeppert et al., 1984) and
potentiometric analysis (Thermo Scientific™ Orion Star AIII). This involved measuring the change in pH of the acetic acid
extractants, relative to CaCO$_3$ standard extracts, which had been mixed with analytical-grade quartz and vortexed for 10
seconds (Rowley et al., 2020). A 2 g portion of ground soil were combined with 25 mL 0.4 $M$ acetic acid and shaken at 250
rpm for 16 h on a rotary shaker. Soil organic C and total N contents was quantified using an elemental analyser
(ThermoScientific FLASH 2000). Inorganic C was removed from the samples through fumigation techniques (Harris et al.,
2001). An inhouse reference soil was measured throughout the runs to check for reproducibility (<5 % relative standard
deviation).



### 2.5 Soil DNA extraction and *frc* gene sequencing

#### 2.5.1 Soil DNA extraction

The abundance and diversity of oxalotrophic microorganisms were assessed via HTS of a fragment of the *frc* gene (Hervé et al., 2018; Khammar et al., 2009). The *frc* gene encodes the formyl coenzyme A (CoA) transferase (EC: 2.8.3.16), a key enzyme in oxalate metabolism and has been shown to be a valuable marker for the identification of oxalotrophs (Khammar et al., 2009). RNAlater was removed from samples by centrifugation and the resultant supernatant discarded. DNA extraction was carried out using the MP FastDNA SPIN Kit (MP Biomedicals) following the kit instructions. DNA

concentration in the resulting extracts was measured using the Qubit™ dsDNA BR Assay kit (Thermo Fisher Scientific). DNA extracts were stored at -20°C until further analyses.

Prior to HTS, the presence of the *frc* gene in the DNA extracts was assessed by polymerase chain reaction (PCR) of a short fragment (155 bp) of the *frc* gene with primers frc171-F (see S.Methods for primer sequences) and frc306-R (Khammar et al., 2009). The positive control was obtained from a colony PCR of the oxalotrophic bacterium *Cupriavidus oxalaticus*

(Palmieri et al., 2022). The PCR was performed using a Thermo Fischer Scientific™ Arktik Thermal Cycler (see S.Methods). Of the initial 45 samples, 31 were selected for sequencing. These included 26 samples that showed an amplification for the *frc* gene and 5 that did not, which were included as control samples (S.Table 3). Samples were sent to Fasteris SA (Geneva) for library preparation and Illumina 2 x 150 bp MiSeq sequencing of a 473 bp fragment of the *frc* gene, obtained with primers frc171-F and frc627-R (Khammar et al., 2009).

#### 235 2.5.2 Sequence analyses

For sequences analysis, the development of a new pipeline was required as the *frc* gene is not a standard marker in microbial ecology (see S.Methods for more details on the procedure). Briefly, raw sequencing reads underwent quality assessment using *FastQC* (Andrews, 2010). Reads were accordingly trimmed using *Trimmomatic* (Bolger et al., 2014), which use specific parameters to remove low-quality bases and short sequences, yielding a high-quality dataset. Subsequently, *LotuS2*

(Özkurt et al., 2022) implementing DADA2 (Callahan et al., 2016) was employed for the identification of unique amplicon sequence variants (ASVs). As there is no public reference database for the *frc* gene target, taxonomic assignment of ASVs was achieved using *RAPPAS* (Linard et al., 2019). For this, a custom phylo-kmer database was constructed from filtered and aligned Formyl-CoA:oxalate CoA-transferase amino acid sequences retrieved from the UniProtKB database (UniProt Consortium 2023; query date: 9th Nov. 2023). Representative ASV sequences were translated, filtered for completeness, and

their corresponding amino acid sequences were then phylogenetically placed using *RAPPAS*. The resulting placement data, along with ASV counts, were then integrated into a *phyloseq* object in R (McMurdie and Holmes, 2013). This enabled a refined taxonomic assignment based on phylogenetic placement likelihood and lowest common ancestor analysis within subtrees, ultimately classifying 30,345 ASVs within the Bacteria kingdom.





### 2.6 Statistical analysis

Analysis and visualisation of the data were conducted in R (RCoreTeam, 2024). Sequencing data were analyzed and plotted for visualisation with the *vegan* package (Oksanen et al., 2024). To enhance clarity in the heatmap visualisation, rare taxa were filtered out by removing genera for which maximum relative abundance was <1 %. Following this, hierarchical clustering of samples was performed using complete linkage (full dataset) or average linkage (filtered/abundant genera) and the Bray-Curtis distance, which generated a column dendrogram for genus organisation within the heatmap. Additionally,

alpha and beta diversity analyses were conducted and are covered in the S.Methods section. Non-metric Multidimensional Scaling (NMDS) analysis was then performed on taxon composition with $k$=2 dimensions. To statistically test for significant differences in community composition between sample types, a permutational multivariate analysis of variance (PERMANOVA) was computed using the adonis2 function in the *vegan* package with 1000 permutations.

Soil and vegetation plots were generated using the *ggplot2* package (Wickham, 2016). Alpha diversity metrics, plant, and

260 soil variables were analyzed for normality using the Shapiro-Wilk test, which indicated that the data did not follow a normal distribution. Consequently, non-parametric tests were applied throughout the study. Kruskal-Wallis tests were used to assess significant differences in median values among independent groups (soil sample type, organic matter type, species, and tree size categorized by DBH). For significant results, the Wilcoxon rank-sum tests (Mann-Whitney U tests) were used for pairwise comparisons to identify specific group differences. To account for multiple comparisons, $p$-values were adjusted

using the Benjamini-Hochberg method, to control the false discovery rate. All statistical tests were performed with a significance threshold of $\alpha$=0.05, and mean values for statistically relevant groupings are presented with ±1 standard deviation.

## 3 Results

### 3.1 Field observations

Field observations of the studied individuals are summarised in S.Table 2. Biomass values calculated using the published allometric equations were skewed by variations in DBH. Consequently, conversion of contents to absolute quantities based upon these biomass values are only reported in the supplementary information. The highest allometrically-calculated biomass was for a *L. tetraphylla* individual (L4=3,688 kg), while one *S. emarginatus* individual had the lowest biomass (S2=90 kg). The biomass of mature tree species differed in the following order: *L. tetraphylla* > *S. emarginatus* ≥ *A.*

*heterophyllus* > *D. ebenum*. All selected species exhibited effervescence in response to 10 % HCl, indicating the presence of $CaCO_3$ (S.Fig. 1). Particularly, strong reactions were observed on the trunks of *S. emarginatus* (S3) and *L. tetraphylla* (L4), but also in the rhizosphere of L4.



## 3.2 Microscopy of biomass

Calcium oxalate crystals of different morphology were observed and identified in all organic matter types. Light microscopy
performed directly at the field site led to the identification of CaOx in 21 out of the 25 tested TDEF species (S.Table 4), thus
highlighting its widespread abundance in the local TDEF ecosystem. CaOx monohydrate crystals (whewellite; $CaC_2O_4.H_2O$)
were observed in all samples of the four selected species as simple or twinned druse (Fig. 2A & B) and prismatic crystals
(Fig. 2C to E), which were the most observed crystal habits. They also were identified in all the tissue types of the different
species. Druse CaOx monohydrate crystals were predominantly identified in leaves and litter (Fig. 2A & B) but not in the
bark of the tree species. CaOx dihydrate (weddellite; $CaC_2O_4.2H_2O$) was rarely detected in *L. tetraphylla* branches and *A.
heterophyllus* bark samples. Opal and echinate sphere phytoliths were also occasionally observed in *A. heterophyllus* leaf
samples. In accordance with field observations, micritic $CaCO_3$ was readily identified in *S. emarginatus* trunk samples (Fig.
2E) and *L. tetraphylla* trunk and root samples (Fig. 2F). Less common forms of $CaCO_3$, including needle-fibre and spheric
$CaCO_3$, were observed in *S. emarginatus* trunk. Overall, biominerals, such as CaOx and $CaCO_3$, were identified in
abundance throughout the different sampled organic matter types of the observed species (Fig. 2 & S.Fig. 2A & B).



A

10 μm

B

200 μm

C

10 μm

D

50 μm

E

10 μm

F

50 μm





**Figure 2. Scanning electron microscopy of biominerals observed in the plants. (A) Druse crystal of CaOx monohydrate in the litter of *Sapindus emarginatus*. (B) CaOx monohydrate druse crystals in a branch of *Artocarpus heterophyllus*. (C) CaOx monohydrate**
**prismatic crystal in *Lepisanthes tetraphylla* bark. (D) CaOx monohydrate prismatic crystals in *Diospyros ebenum* bark. (E) CaOx monohydrate prismatic crystal surrounded by micritic CaCO₃ deposits in the trunk wood of *S. emarginatus*. (F) Aggregated calcified root cells in *L. tetraphylla*. More micrographs and spectroscopy of observed crystals are provided in S.Fig. 2.**

### 3.3 Calcium oxalate content of biomass and soils

The enzymatic oxalate kits revealed that, across all plant samples, the average CaOx content of the different organic matter
types was 4.4±3.2 % D.W. (S.Table 5). Although CaOx was measured in soils from the hollowed-out trunk of S3 (336 mg kg$^{-1}$) and D3 (40 mg kg$^{-1}$), CaOx was not detected in other soil samples. Among plant tissues, bark samples exhibited higher CaOx contents (7.83±4.6 % D.W.), with contents decreasing sequentially as follows: bark > leaves ≥ branches ≥ litter ≥ roots (Fig. 3A). Significantly higher CaOx content (3.42±1.14 % D.W.; Wilcoxon test, $p$-value=0.01) were observed in smaller trees (<30 cm DBH) compared to larger individuals (>30 cm DBH; 2.69±1.06 %
D.W.). At the species level, the bark of *D. ebenum* displayed a significantly higher CaOx content (13.4±3.29 % D.W.) than other organic matter types from the same species. In *A. heterophyllus*, branches (5.82±2.37 % D.W.) and bark (4.91±1.92 % D.W.) also had higher CaOx content than other sample types from the same species (Fig. 3B; S.Table 5).

The absolute quantity of CaOx in each species, based on the allometrically-calculated biomass and the proportion of the organic matter types, followed the same order as the sampled trees' biomass: *L. tetraphylla* > *A. heterophyllus* > *S.*
*emarginatus* > *D. ebenum* (S.Fig. 3 & S.Table 5).

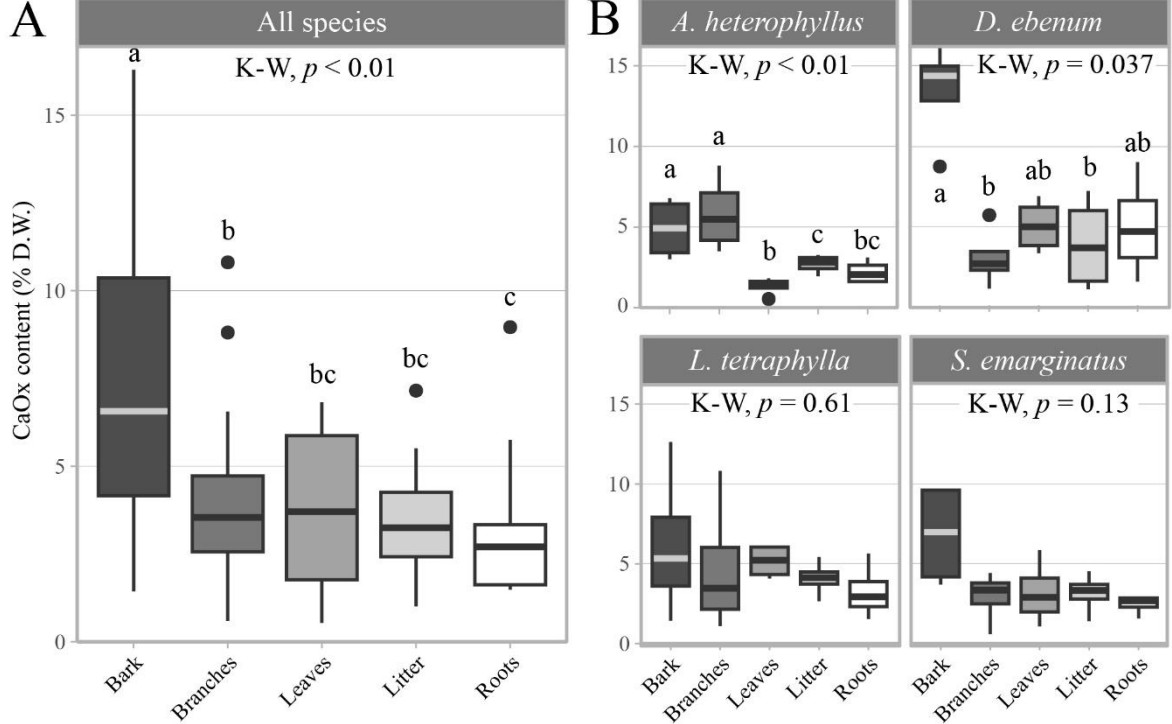





**Figure 3. Calcium oxalate (CaOx) contents within the plant tissues. (A) CaOx contents averaged over all tree species (% dry weight of CaC$_2$O$_4$.H$_2$O). (B) Species specific organic matter CaOx contents:** *Artocarpus heterophyllus, Diospyros ebenum, Lepisanthes tetraphylla*, **and** *Sapindus emarginatus*. **The midline in the box plot represents the median, box extents are the 1$^{st}$ and 3$^{rd}$ quartiles, and whiskers reach 1.5 times the inter-quartile range, while the dots represent outliers, outside of this range. Significant differences between groups were assessed using the Kruskal-Wallis (K-W) test followed by the Wilcoxon rank-sum test adjusted Benjamini-Hochberg (BH) method, and letters displayed above boxplots refer to significantly different groups ($\alpha$=0.05).**

### 3.4 Microbiological analysis - *frc*-gene high-throughput sequencing

From the initial 31,320 ASV counts obtained from *LotuS2*, 30,345 were assigned to the Bacteria kingdom, meaning that only 3 % were not assigned to a group. This supports the fact that the region used is prevalently found in this kingdom. In total, 86 % of these bacterial ASVs were assigned to Actinomycetota and 13.8 % to Pseudomonadota (S.Fig. 4 & 5). The most prevalent taxon in the dataset, irrespective of sample type, was an agglomerate of Streptosporangiaceae (uncultured), belonging to the Actinomycetota phylum, with approximately 50 % relative abundance (S.Fig. 6). Furthermore, the most frequent observed genera in the whole dataset were *Saccharopolyspora, Nonomuraea, Planotetraspora, Planobispora,* and *Kutzneria* (Table 1; S.Fig. 6).

**Table 1. Genera with the highest relative abundance from *frc* sequencing analysis across all analyzed tree species and sample types, ordered by their relative abundance (%) in each sample type. Top ten genera for each sample type are marked in bold.**

| Genus | Phylum | Litter | Roots | Rhizosphere soil | Adjacent soil | Control soil |
|---|---|---|---|---|---|---|
| Streptosporangiaceae uncul. | Actinomycetota | **54.6** | **46.9** | **51.9** | **51.4** | **45.4** |
| *Saccharopolyspora* | Actinomycetota | **5.7** | **9.8** | **19** | **19.3** | **20.9** |
| Rhodocyclales uncul. | Pseudomonadota | 0.2 | **8.9** | 0.2 | 0.1 | 0.1 |
| *Nonomuraea* | Actinomycetota | **5.2** | **8.9** | **4.8** | **4.1** | **4.2** |
| *Planotetraspora* | Actinomycetota | **3** | **3.9** | **5** | **4** | **4.9** |
| *Methylobrevis* | Pseudomonadota | **4.5** | 0.9 | 0.5 | 0.1 | 0.1 |
| *Planobispora* | Actinomycetota | **1.8** | **3.1** | **3.4** | **3.6** | **4.4** |
| *Kutzneria* | Actinomycetota | 0.8 | **2.9** | **1.8** | **3.6** | **1.6** |
| *Aromatoleum* | Pseudomonadota | 0.2 | 0.3 | 0.7 | **3.5** | **3.1** |
| *Xanthobacter* | Pseudomonadota | **2.8** | 0 | 0 | 0 | 0 |
| *Methylorubrum* | Pseudomonadota | **2.5** | 0.2 | 0.1 | 0.1 | 0.2 |
| *Mycolicibacterium* | Actinomycetota | 1.3 | 0.7 | **2.1** | **2** | 1.2 |
| *Actinosynnema* | Actinomycetota | 0.6 | 0.4 | **1.9** | **1.9** | **1.4** |
| *Tardiphaga* | Pseudomonadota | **1.9** | 0 | 0.1 | 0 | 0 |
| *Methylobacterium* | Pseudomonadota | **1.9** | 0.2 | 0.3 | 0.2 | 0.1 |
| *Propylenella* | Pseudomonadota | 0 | **1.8** | 0.5 | 0.7 | 0.3 |
| *Actinacidiphila* | Actinomycetota | 0.3 | **1.1** | **1.3** | 0.6 | **1.8** |
| *Bradyrhizobium* | Pseudomonadota | 1.2 | **1.4** | 0.7 | 0.3 | **1.4** |



| *Pigmentiphaga* | Pseudomonadota | 0.7 | 0.9 | **0.8** | **1** | 0.8 |
|---|---|---|---|---|---|---|

Root and adjacent soil samples exhibited the lowest median alpha diversity values (Shannon and Simpson) and a pairwise
Wilcoxon rank-sum tests revealed that there was a significant difference only between root and litter samples (*p*-value=0.02; S.Fig. 7A). Control soils displayed the highest median value in the Shannon diversity index, with a significant difference observed solely between root and control soil samples (*p*-value=0.03; S.Fig. 7B). The Simpson diversity index showed relatively stable median values across sample types, and again, only root and control soil samples presented a significant difference (*p*-value=0.03; S.Fig. 7C). The NMDS analysis of taxon lists revealed clear clustering of samples by tree species
and sample types (S.Fig. 8) and the PERMANOVA analysis indicated that all pairwise comparisons of samples by types or tree species were significantly different (S.Table 6).

The *frc* gene sequence analysis is summarised in a heatmap (Fig. 4), which highlights the distribution of taxa as a function of sample type. In this analysis, an unclassified genus of the family "Streptosporangiaceae uncul." was removed to enhance figure clarity as it consistently represented ~50 % of the ASVs across different sample types (Table 1), thereby dominating
the heatmap colour-scale. The analysis showed that litter samples clustered together, with different representative taxa for *L. tetraphylla* as compared to *D. ebenum* and *S. emarginatus*, supporting the NMDS analysis (S.Fig. 8). Litter samples from *L. tetraphylla* were associated to *Micromonospora*, uncultured Burkholderiales, *Methylobrevis*, and *Tardiphaga*. On the other hand, *D. ebenum* and *S. emarginatus* litter samples were associated to *Xanthobacter*, *Hartmannibacter*, *Hansschlegelia*, *Methylorubrum*, and *Methylobacterium*. Then, soil samples showed characteristic taxa, with *Saccharopolyspora* present at
higher relative abundances compared to other sample types. Interestingly, control soil samples (except for one sample) were found to be uniquely characterized by a specific assemblage of putative oxalotrophs, the genera *Afipia* and *Nitrospira*. Finally, *Pandoraea*, *Rhabdonatronobacter*, and an uncultured Rhodocyclales were specifically associated with roots samples, while *Methylobrevis* was largely associated with roots, along with the litter of *L. tetraphylla* (Fig. 4).




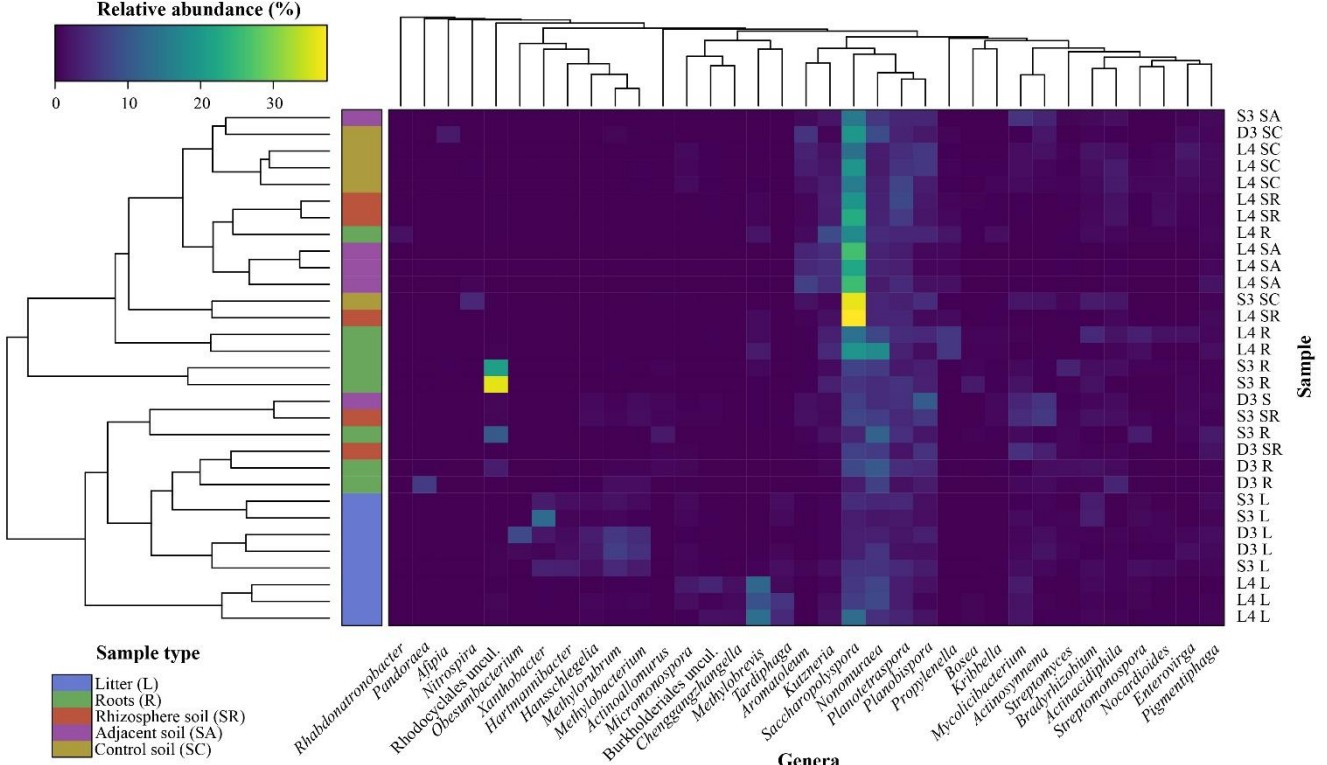

Figure 4. **Heatmap of the Bray-Curtis clustering analysis of *frc* gene sequencing results with a column dendrogram, which organizes the genera and sample types, and the statistical similarities amongst them. Sample names correspond to the trees sampled (D3 - *Diospyros ebenum*, L4 - *Lepisanthes tetraphylla*, and S3 - *Sapindus emarginatus*) and the sample type (SC - control soil, SA - adjacent soil, SR - rhizosphere soil, R - roots, and L - litter). The uncultured genus of the family "Streptosporangiaceae uncul." was removed to enhance figure clarity as it consistently represented ~50 % of the amplicon sequence variants. Similarly, rare taxa were filtered out by removing genera for which maximum relative abundance was <1 %.**

### 3.5 Electron microscopy of microbial features

Scanning electron microscopy enabled the imaging of microorganisms actively interacting with CaOx crystals in the organic matter of the different tree species. Characteristic morphological features, such as cell dimensions, filamentation, and branching patterns, were used to identify the potential groups of organisms that the micrographs illustrated. A clustered form (~20 μm in diameter) of mycelia-forming Actinomycetota (hypha diameter ~1 μm) was identified in branch and bark samples from *A. heterophyllus* (Fig. 5A). This microorganism was imaged degrading CaOx monohydrate crystals in the *A. heterophyllus* bark (Fig. 5B), coating themselves with a mixture of primary CaOx and secondary $CaCO_3$ crystals. Actinomycetota spores were also observed in association with CaOx monohydrate crystals in *D. ebenum* trunk samples (Fig. 5C). Finally, fungal hyphae (hypha width ~2.5 μm supporting a fungal nature) were also observed in *L. tetraphylla* root sample (Fig. 5D). The possible presence of an anastomosis clamp indicates that, at least, the fungal hyphae in Fig. 5D likely belong to a Basidiomycota fungus. Overall micrographs supported the *frc* gene sequencing results, suggesting that Actinomycetota are an important phylum of oxalotrophic organisms in the sampled tree species.





**Figure 5. Micrographs of microbial organisms associated with calcium oxalate (CaOx) crystals. (A) Coated, mycelia-forming Actinomycetota clusters in *Artocarpus heterophyllus* bark. (B) Coated, mycelia-forming Actinomycetota clusters capable of metabolising CaOx crystals in *Artocarpus heterophyllus* bark. (C) Actinomycetota spores in a soil taken from the trunk of *Diospyros ebenum.* (D) Broad (Basidiomycota) and narrow (Actinomycetota) hyphae associated with prismatic CaOx crystals in the root material of *Lepisanthes tetraphylla.***

### 3.6 Soil chemical properties

Bulk soil properties (S.Table 7), total element (S.Table 8) and mineral contents (S.Table 9) are reported in the Supplementary Materials. There were small differences in individual soil chemical properties between the sites, but nothing that differentiated the sites based specifically on parent material (S.Fig. 10 & 11). Soil $pH_{KCl}$ was slightly higher in the Cuddalore sandstone (5.4±0.7) than in the Alluvium (4.8±0.6), although the difference was not statistically significant (S.Fig. 9). Soil pH values were higher in the hollowed-out trunks (7.4±0.6) than in the adjacent (5.2±0.6) and the control (5.1±0.7) soils (Fig. 6A), but the adjacent and control soils displayed no statistically significant difference. The cation




exchange capacity was also higher in the hollowed-out trunks (19.8±4.0 cmol$_c$.kg$^{-1}$), compared to the adjacent (4.0±3.1 cmol$_c$.kg$^{-1}$) and the control (4.2±4.9 cmol$_c$.kg$^{-1}$) soils (Fig. 6B). Na$_{Exch}$ remained below the detection limit in all samples. This difference in cation exchange capacity was mainly driven by a higher Ca$_{Exch}$ content (Fig. 6D), but also by Mg$_{Exch}$ (Fig. 6E), and K$_{Exch}$ (Fig. 6F). All 3 of the major cations were higher in the trunk (Ca$_{Exch}$=13.5±3.2 cmol$_c$.kg$^{-1}$; Mg$_{Exch}$=3.9±1.7

385    cmol$_c$.kg$^{-1}$; K$_{Exch}$=2.4±2.1 cmol$_c$.kg$^{-1}$) than in the adjacent (Ca$_{Exch}$=2.8±2.4 cmol$_c$.kg$^{-1}$; Mg$_{Exch}$=0.9±0.7 cmol$_c$.kg$^{-1}$; K$_{Exch}$=0.2±0.1 cmol$_c$.kg$^{-1}$) or the control soils (Ca$_{Exch}$=3.1±3.9 cmol$_c$.kg$^{-1}$; Mg$_{Exch}$=0.9±1.0 cmol$_c$.kg$^{-1}$; K$_{Exch}$=0.1±0.1 cmol$_c$.kg$^{-1}$). While the differences were not significant ($p$-value=0.09), Al$_{Extr}$ showed an opposite trend, decreasing in content moving from the control > adjacent > trunk soils (Fig. 6C). Overall, there were limited differences between the sampled adjacent and control soils, but there were distinct shifts in soil chemical properties of the hollowed-out trunk environment of

the TDEF species (*D. ebenum*, *L. tetraphylla*, and *S. emarginatus*).

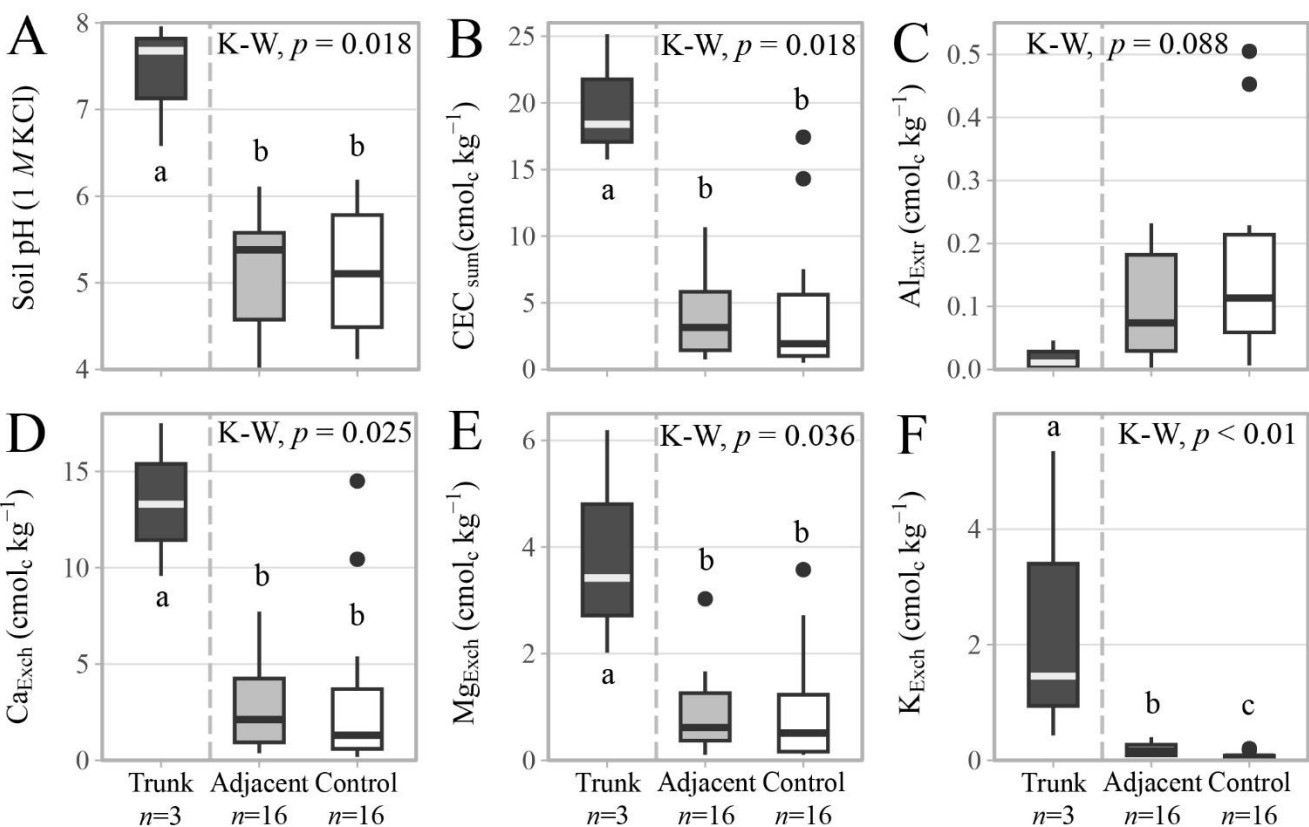

**Figure 6. Soil chemical properties in the hollowed-out trunks (*n*=3), adjacent (*n*=16) and control soil samples (*n*=16) for all tree species across the different parent materials. (A) Soil pH measured in 1 *M* potassium chloride. (B) Cation exchange capacity calculated as the sum of extracted exchangeable cations (centimoles of charge kg$^{-1}$). (C) Extractable aluminium content. (D)**

**Exchangeable calcium content. (E) Exchangeable magnesium content. (F) Exchangeable potassium content. Symbols and box plots as in Fig. 3. See S.Fig. 9 to 11 for differences between parent materials or S.Table 7 for the rest of the bulk soil properties.**




Soil organic C contents were higher in the trunks (42.8±18.6 g kg$^{-1}$) and adjacent soils (8.0±1.5 g kg$^{-1}$) than the control soils (4.9±0.5 g kg$^{-1}$), while the total nitrogen contents were higher in the trunk soils (3.3±1.3 g kg$^{-1}$), but similar between the adjacent (0.6±0.1 g kg$^{-1}$) and control soils (0.4±<0.0 g kg$^{-1}$). A trend for C:N ratios to decrease with distance from the tree

was observed, moving from the hollowed-out trunk samples (13.7±0.4) to the adjacent (12.8±0.5) and control soil samples (12.3±0.8), but the differences were not statistically significant (*p*-value=0.3). Meanwhile, the CaCO$_3$ contents were higher in the hollowed-out trunks (29.1±11.3 g kg$^{-1}$) than in the adjacent and the control soils, which had a far lower CaCO$_3$ content that was near the limits of detection with this method and did not show a statistically significant difference. Thus, while there was an increase in soil CaCO$_3$ content within the hollowed-out CaCO$_3$-coated trunk environment of the trees, there was very

little CaCO$_3$ in the adjacent surficial soil samples, which displayed no significant difference relative to the control soils.

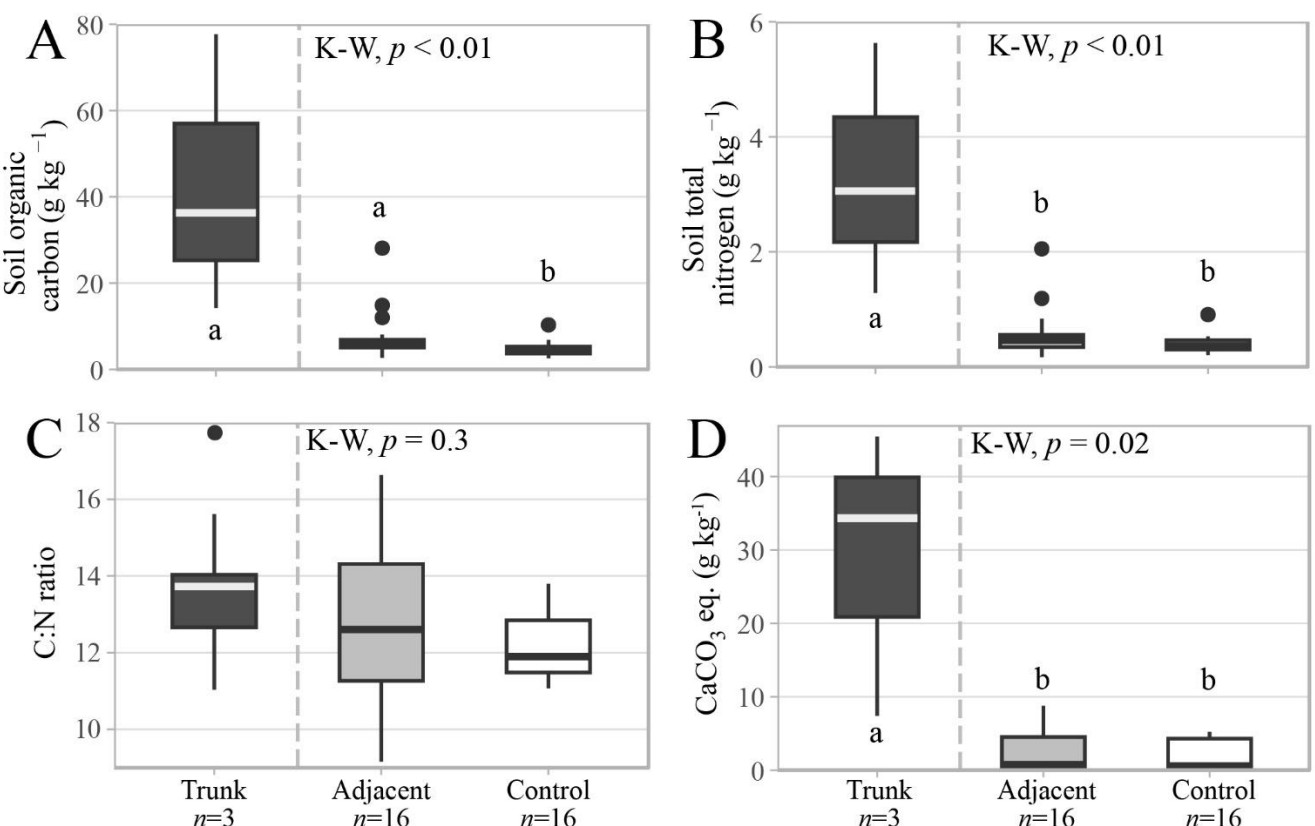

**Figure 7. Soil carbon and nitrogen contents in the hollowed-out trunks (*n*=3), adjacent (*n*=16) and control soil samples (*n*=16) for all sampled species. (A) Soil organic carbon content (g kg$^{-1}$). (B) Soil total nitrogen content (g kg$^{-1}$). (C) Carbon to nitrogen ratio. (D) Calcium carbonate equivalent content (g kg$^{-1}$). Symbols and box plots as in Fig. 3.**





## 4 Discussion

### 4.1 Calcium oxalate forms and functions

The study identified four active and novel OCP ecosystems associated with three diagnostic TDEF species and one local agroforestry species, all of which produced significant quantities of CaOx. Although CaOx was not readily identified in the tree-adjacent soils, CaOx crystals were abundant across the sampled species in all analyzed organic matter types, including in litter around the trees. No statistically significant differences were found between tree species, but smaller trees exhibited proportionally higher amounts of CaOx than larger individuals, consistent with other non-woody plants (Jones and Ford, 1972; Rahman and Kawamura, 2011), and *Brosimum alicastrum* Sw. (family Moraceae), another species with an active OCP (Álvarez-Rivera et al., 2021; Rowley et al., 2017). In young plants, CaOx is thought to protect developing cells from high cytoplasmic $Ca^{2+}$ concentrations, which can hinder cell expansion and growth. However, CaOx crystals can also contribute to defence of plants against herbivory (Franceschi and Nakata, 2005; Hudgins et al., 2003; Ward et al., 1997) and CaOx content was higher in external tissues, such as bark, which are more prone to encounter predation. Raphide (needle-like) CaOx crystals are typically associated with herbivore defence (Lawrie et al., 2023), and were not extensively observed in microscopy data, but prismatic crystals have also been reported to contribute to defensive functions in plants (Korth et al., 2006). As younger trees are more vulnerable to herbivory, we hypothesise that they likely invest more energy in producing CaOx as a defensive strategy (Ward et al., 1997; Wiggins et al., 2016), which would explain the higher CaOx contents observed in the biomass of smaller individuals in the TDEF. The widespread production of CaOx observed in the four studied tree species, as well as in a broader range of local species, highlights its significant, yet overlooked role in the Indian TDEF; however, its absence in the adjacent soils points to a disconnect between its aboveground production and belowground persistence.

### 4.2 Oxalotrophic bacteria communities in the TDEF

The absence of CaOx in the adjacent soils could be explained by the oxalotrophic microbial community, which was characterized with HTS of the *frc* gene (coding for a Formyl-CoA:oxalate CoA-transferase; Khammar et al. 2009). Prior to discussing the environmental relevance of the results, key assumptions of using the *frc* gene as a marker of oxalotrophy should be acknowledged (Sonke and Trembath-Reichert, 2023). The first assumption is that the enzyme encoded by the *frc* gene, Formyl-CoA:oxalate CoA-transferase, establishes a direct metabolic link between formate, oxalate, and oxalotrophy. This enzyme plays a key role in the oxalate degradation pathway of many oxalotrophic bacteria (Anantharam et al., 1989; Ruan et al., 1992; Sahin, 2003), such as the obligate oxalotroph *Oxalobacter formigenes* (Ruan et al., 1992). Yet, there are currently six biological oxalate degradation pathways documented, and not all of them rely on Formyl-CoA transferase (Sahin, 2003; Schneider et al., 2012). An example of which, *Methylobacterium extorquens* (Schneider et al., 2012) can convert oxalate to glyoxylate using Oxalyl-CoA reductase. Conversely, not all organisms harbouring the *frc* gene are actual



oxalotrophs as this gene appears to be associated to several other metabolisms (Sonke and Trembath-Reichert, 2023). As a result, an analysis based on the *frc* gene can only provide a partial image of the oxalotrophic guild.

Another key issue with investigating less commonly studied genes, such as the *frc* gene, in environmentally complex samples is the limited availability of reference databases for taxonomic annotation. For this reason, in this study, a database
had to be created to enable accurate identification of the oxalotrophic guild. While phylogenetic placement offers a work around to this constraint (Czech et al., 2022), a significant limitation arises from the inherent dependence of the placement on the breadth of available reference sequences in databases. Taxonomic resolution hinges on the alignment's diversity and annotation quality, although it can be limited by uneven clade representation and the gene's sporadic evolution via horizontal transfer or phylogenetic restriction (Czech et al., 2022). Hence, due to these assumptions and limitations, HTS of the *frc* gene
provides a partial and potentially biased snapshot of the oxalotrophic guild in the studied samples. Nevertheless, HTS of the *frc* gene still represents the best tool available for investigating oxalotrophic microbial communities in complex environmental samples.

Despite these limitations, the HTS analysis identified a range of well-known oxalotrophs (listed in S.Table 10), including the order Burkholderiales, which contains *Oxalobacter formigenes*, as well as non-obligate species belonging to the genera
*Burkholderia* (Kost et al., 2014), *Cupriavidus* (Palmieri et al., 2022), *Ralstonia* (Sahin, 2003), and *Pandoraea* (Sahin et al., 2011). Other recognized oxalotrophic genera identified were *Xanthobacter* (Sahin et al., 2002) and *Afipia* (Bravo et al., 2015), the latter identified only in a control soil sample. The identification of oxalotrophs in control samples is unsurprising given that most oxalotrophs are not obligate and use CaOx as an alternative C source (Hervé et al., 2016). Thus, oxalotrophs highlighted in control soils, far from a CaOx source, may not be actively metabolising oxalate. The higher alpha-diversity
indices in control soil samples could be linked to the lower selective pressure on bacterial communities located away from oxalogenic trees. This is further supported by beta-diversity analysis (PERMANOVA), which showed that all sample types and trees had significatively different oxalotrophic microbial communities. The detection of known oxalotrophs via HTS of the *frc* gene supports its utility in partially characterizing oxalotrophic microbial communities and highlights that communities in the TDEF are structured and closely linked to tree-derived oxalate inputs.

In addition to known oxalotrophs, the HTS also identified several taxa that have not previously been described as oxalotrophs. For instance, members of the Streptosporangiaceae family were detected, including the genera *Nonomuraea*, *Planotetraspora*, and *Planobispora*, a grouping of aerobic, Gram-positive, chemoorganotrophic Actinomycetota that form a branched, non-fragmenting mycelium (Otoguro et al., 2014). Likewise, genera within the Pseudonocardiaceae family not previously described as oxalotrophs include *Kutzneria* or *Saccharopolyspora* (Wei et al., 2023), the latter containing a
species shown to be unable to decarboxylate oxalate (Meklat et al., 2014). Other taxa not previously associated with oxalotrophy included *Methylobrevis*, a methylotrophic genus (Poroshina et al., 2015), and *Hansschlegelia*, from the Methylocystaceae family known for methane oxidation via the serine pathway (Webb et al., 2014). *Nitrospira*, a chemolithoautotrophic nitrite-oxidising genus (Bayer et al., 2021) important for nitrification (Daims and Wagner, 2018) was also identified and has been previously shown to grow on formate as a C source, which may explain its presence in our





dataset. Nitrogen-cycling taxa, including *Hartmannibacter* (Suarez et al., 2014) and uncultured Rhodocyclales (Ding et al., 2022; Liang et al., 2023) were also detected by the HTS analysis, but their oxalotrophic potential remains unclear. Ultimately, for all taxa not previously confirmed as oxalotrophs, their role in oxalate utilisation would require further investigation. Some assignments may represent false positives, i.e. taxa harbouring the *frc* gene but using it in another metabolic pathway (Sonke and Trembath-Reichert, 2023), while others may represent true oxalotrophs, highlighting that the

diversity of oxalotrophic bacteria remains largely unexplored.

Overall, the HTS of the *frc* gene highlighted that the most prominent taxonomic phylum was Actinomycetota (86 %). This was congruent with separate SEM EDX observations in the bark of *A. heterophyllus*, which directly imaged mycelia-forming Actinomycetota associated with CaOx (Fig. 5A & B). Actinomycetota were identified in all samples and contain well-known oxalotrophic taxa (Bravo et al., 2013; Robertson and Meyers, 2022; Sonke and Trembath-Reichert, 2023), which, along with

Pseudomonadota, are among the most prominent taxonomic groups involved in oxalate metabolism, both in the soil (Actinomycetota) or on plants (Pseudomonadota; Hervé et al. 2016). To conclude, oxalotrophs, predominated by Actinomycetota, were highlighted using HTS of the *frc* gene and were also imaged directly interacting with CaOx. This suggests that the absence of CaOx in the adjacent soils, despite its widespread production in the TDEF, is likely due to its rapid metabolism by an active oxalotrophic community associated with the investigated tree species in which

Actinomycetota play a key role.

## 4.3 Inorganic carbon precipitation: the biogenic reversal of pedogenic trajectories

Although only three hollowed-out trunk soils were analyzed, each presented clear biogeochemical evidence of an active OCP. Notable observations included a localised (relative to the control) increase in soil pH (+1.3 pH units), elevated $Ca_{Exch}$ content (+8.2 $cmol_c.kg^{-1}$), and precipitation of $CaCO_3$, both on the trees, and within the hollowed trunk soil (+23 g $kg^{-1}$

$CaCO_3$ equivalent). Similarly, Pons et al. (2018) reported a $pH_{H2O}$ increase of +1.5 pH units at the foot of *M. excelsa*, compared to control soils located 30 m away. Yet, it has also been demonstrated by Cailleau et al. (2004) that *M. excelsa* is linked with far greater increases in $CaCO_3$ (+300 g $kg^{-1}$ $CaCO_3$ relative to control soils) than reported in this study. The difference in $CaCO_3$ content between these studies could partially be attributed to the larger biomass of *M. excelsa*, which is an emergent canopy tree (~40 m height) that produces far greater absolute quantities of CaOx, compared to the smaller

TDEF trees. These comparisons underscore how tree functional traits, particularly size, biomass, and CaOx production capacity can modulate the strength of OCP-driven changes in different forest systems.

Beyond classical measures of an active OCP, a decreasing trend in extractable $Al^{3+}$ contents was seen, moving from the control soils to the hollowed-out trunk environments of trees. It is well established in soil science that more-soluble cations ($Ca_{Exch}$) and easily-weathered minerals ($CaCO_3$) typically decrease in content with pedogenesis, time, increased rate of

weathering, and a higher or positive local water budget (balance between precipitation and evapotranspiration; Jenny 1941; Doetterl et al. 2025). Meanwhile, less mobile ions and minerals, such as $Al^{3+}$ and oxides remain because they are relatively more resistant to weathering (Adams et al., 2000; Bloom et al., 1979; Reuss et al., 1990). An example of this on a global



scale, Slessarev et al. (2016) demonstrated that soil pH is clustered into two separate groups by water budget, governed by $CaCO_3$ equilibria in drier environments (pH ~8.2) or gibbsite/$Al^{3+}$ hydrolysis in wetter environments (pH ~5.1). Thus, as pedogenesis progresses, soluble minerals and cations such as $CaCO_3$ are leached from humid soil environments, shifting the soil from a pH range governed by $CaCO_3$ equilibria to a system dominated by gibbsite/$Al^{3+}$ hydrolysis, increasing $Al^{3+}$ saturation (Chadwick and Chorover, 2001; Rowley et al., 2020; Slessarev et al., 2016). However, an active OCP can act to biogenically reverse this conventional model of pedogenesis on a local scale as the process drives alkalinisation and the precipitation of relatively soluble cations/minerals (Fig. 8C). The observed decline in extractable $Al^{3+}$ from control soils to tree trunks may reflect this shift, indicating a transition away from $Al^{3+}$-dominated systems towards Ca-rich, biologically-influenced $CaCO_3$ equilibria, consistent with localised OCP activity. Yet, the absence of large differences between the biogeochemical properties of adjacent and control soils still requires explanation.

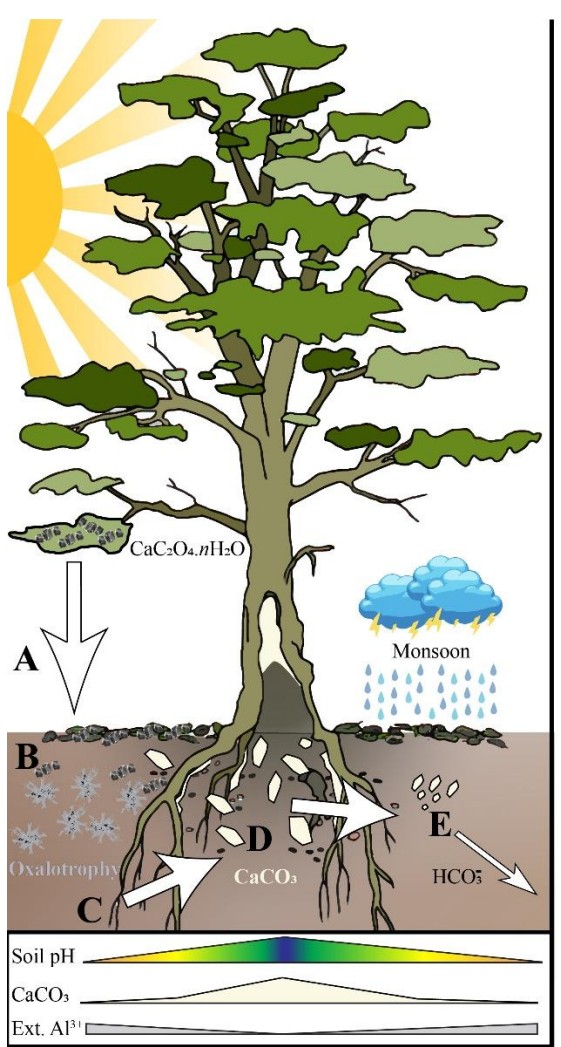

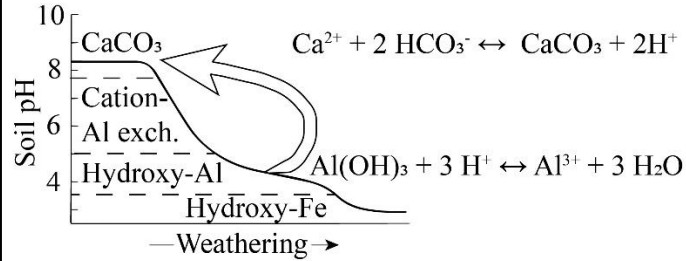

**A = Oxalate biosynthesis during photosynthesis**

$1000\ Ca^{2+} + 1000\ C_2O_4^{2-} + nH_2O = 1000\ CaC_2O_4.nH_2O$

**B = Oxalotrophy**

$1000\ CaC_2O_4.H_2O + 372\ O_2 + 64\ H_2O + 32\ NH_4^+ \rightarrow$
$32\ C_4H_8O_2N^+_{Biomass} + 1872\ HCO_3^- + 936\ Ca^{2+} + 64\ Ca(OH)_2$

**C = Shift in soil buffering system, increasing soil pH**

$Ca^{2+} + 2\ HCO_3^- \leftrightarrow CaCO_3 + 2H^+$

$Al(OH)_3 + 3\ H^+ \leftrightarrow Al^{3+} + 3\ H_2O$

∴ $\sim 43\ Al^{3+} + 128\ H_2O \rightarrow \sim 43\ Al(OH)_3 + 128\ H^+$
$128\ H^+ + 64\ Ca(OH)_2 \rightarrow 64\ Ca^{2+} + 128\ H_2O$

**D = pH increases, precipitating $CaCO_3$**

$936\ Ca^{2+} + 1872\ HCO_3^- \rightarrow 936\ CaCO_3 + 936\ CO_2\uparrow + 936\ H_2O$

**E = Monsoon rain leaches $CaCO_3$**

$936\ H_2O + 936\ CO_2\downarrow \rightarrow 936\ H_2CO_3$
$936\ H_2CO_3 + 936\ CaCO_3 \rightarrow 936\ Ca^{2+} + 1872\ HCO_3^-$



**Figure 8. Conceptual model of the novel oxalate-carbonate pathways and their effects on local soil biogeochemistry in the tropical dry evergreen forests of Tamil Nadu. (A) Calcium oxalate (CaOx) biosynthesis from oxalic acid, produced during photosynthesis (see S.Eq. 1 for more details). Produced crystals are predominantly CaOx monohydrate (CaC$_2$O$_4$.H$_2$O) formed within idioblast cells, which are then released to the surrounding environment. (B) CaOx is metabolised during oxalotrophic activity, releasing HCO$_3^-$ and calcium hydroxide (Ca[OH]$_2$). (C) This process creates an alkalinisation, and in turn, shifts the soil pH buffering mechanism, eventually reversing the typical pedogenic progression induced by weathering, involving the loss of soluble minerals and cations at the expense of resistant minerals or immobile elements (adapted from Chadwick and Chorover 2001). (D) As soil pH increases, CaCO$_3$ begins to precipitate around the trunk and roots as byproducts of oxalotrophy. (E) Carbonate precipitated in the adjacent soils is then leached during monsoon precipitation events.**

### 4.4 What happens to the inorganic C?

While CaCO$_3$ precipitation was abundant both aboveground and within the hollowed-out trunk environments of the TDEF species, there were no statistically significant difference between the adjacent and control soils. This is likely due in part to the fact that only surficial soil layers (0-10 cm) were sampled at random locations, both adjacent to, and at a control distance from the 16 trees. A more extensive sampling campaign was beyond the scope of this initial study, partly because the TDEF forests are considered sacred, despite the absence of formal legal protection. However, incremental sampling at multiple depths, including full soil profiles, focusing on the base of the larger trees, and sampling with distance from the rhizosphere, would likely have yielded starker differences in the adjacent and control soil biogeochemical properties. Indeed, significant CaCO$_3$ precipitation was found associated with the rhizosphere of *L. tetraphylla* during sampling, supporting this conclusion. Consequently, a more detailed analysis of the change in CaCO$_3$ with depth would be required to ascertain the exact C budget of the active TDEF OCP systems and the absence of differences between adjacent and control soils could partially be linked to this study's sampling strategy.

However, these results are consistent with those reported by Hervé et al. (2018), who investigated an active OCP associated with *T. bellirica* in Madhya Pradesh, Central India. The authors found that while CaCO$_3$ accumulated up to 82 % D.W. in bark, there was relatively little CaCO$_3$ in the adjacent soils (+10 g kg$^{-1}$ CaCO$_3$ relative to control soils). The authors hypothesised that this discrepancy between above- and belowground CaCO$_3$ was explained by monsoon precipitation (1100 mm, approximately 75 % of annual precipitation in 4 months), which dissolved and leached OCP-precipitated CaCO$_3$ from the adjacent soils. The present study lends further support to this idea, as while conditions were slightly drier in Tamil Nadu than in Madhya Pradesh (Duraisamy Rajasekaran et al., 2024; Hervé et al., 2018; Saikranthi et al., 2024), the area remains subject to two separate seasons, with dry (inter-monsoons) and wet (monsoons) periods. This hypothesis is further substantiated by the fact that there was undetectable level of Na$_{Exch}$ in the TDEF soils, a soluble monovalent cation with a large hydration sphere, which was likely leached from the system during the rainy periods. Consequently, the low CaCO$_3$ content of soils adjacent to the oxalogenic TDEF trees was likely a result of the region's humid monsoon conditions. This further suggests that a critical water budget threshold may determine whether an active OCP leads to sustained CaCO$_3$ accumulation (Slessarev et al., 2016). However, this threshold is unlikely to depend solely on local water budget but could also be shaped by the intensity and frequency of rainfall events, their interplay with temperature and evapotranspiration, and the resultant impacts on CaCO$_3$ weathering, dissolution, leaching, and reprecipitation dynamics.





**4.5 Can the system still be considered as an inorganic carbon sequestration if its carbonate phase is leached?**

The extent to which OCP systems in the TDEF sequester C will depend on complex interactions between local and regional geochemical reactions (Fig. 8D & E), including (i) the initial source of Ca utilized to sequester $CaCO_3$ biogenically (Fig. 8A & D), (ii) whether it is carbonic acid that weathers OCP-precipitated $CaCO_3$, and (iii) the physicochemical or biogenic processes that determine whether leached $HCO_3^-$ remains in solution or precipitates as a secondary carbonate phases. While local $CaCO_3$ stocks may decrease due to monsoon leaching, the broader-scale sequestration potential remains significant if the inorganic C remains in solution and is exported from the system, instead of precipitating in soils as secondary $CaCO_3$ phases (S.Eq. 2). Percolating soil solutions enriched with dissolved $HCO_3^-$, $CO_3^{2-}$ and $Ca_{Exch}$ will eventually flow out of the TDEF system into the Bay of Bengal, and could represent an important inorganic C sink (Beerling et al., 2020; Taylor et al., 2021; Zamanian et al., 2016). Nevertheless, key uncertainties remain, including the mechanisms that govern Ca and C fluxes and their sensitivity to climatic variables. To address these, isotopic labelling experiments could now be used to precisely trace the cycle of C from CaOx through $CaCO_3$ precipitation, and then its leaching under controlled conditions. This study suggests that the OCP might be globally distributed, potentially existing in more humid environments than previously recognized, and represents an understudied soil inorganic C sequestration process.

**5 Conclusions**

The present study identified novel oxalate-carbonate pathways (OCP) associated with four tree species (*A. heterophylla*, *D. ebenum*, *L. tetraphylla*, and *S. emarginatus*) in the restored and primary tropical dry evergreen forests (TDEF) of Tamil Nadu, India. The trees had an average calcium oxalate (CaOx) content of 4.4±3.2 % D.W, which was higher in the smaller sampled trees, but absolute quantities were higher in larger individuals due to their increased biomass. Well-known oxalotrophic microorganisms were highlighted in all assessed samples by high-throughput sequencing (HTS) of the *frc* gene, including control samples, while Actinomycetota were by far the most represented phylum in all samples (86 %). This was corroborated by microscopy images, which confirmed that filamentous bacteria were directly associated with CaOx monohydrate crystals. Aboveground $CaCO_3$ precipitation was detected in all the tree species, particularly on trunks and wounds on the surface of the trees and was likely linked to oxalotrophy. Large shifts in soil chemistry were observed in the hollowed-out trunks of trees, where there was an increase in local soil pH (+1.3 pH units relative to control), $Ca_{Exch}$ contents (+8.2 $cmol_c.kg^{-1}$), and precipitation of $CaCO_3$ (+23 g $kg^{-1}$ equivalent content). However, the differences between the adjacent and control surficial soils (0-10 cm) were not significant, suggesting that $CaCO_3$ precipitated through the OCP was later leached during the monsoon season in the Indian TDEF ecosystem. These findings underscore the notion that a critical water budget threshold likely governs whether the oxalate-carbonate pathway leads to sustained, local $CaCO_3$ accumulation. Overall, the OCP likely represents an understudied pathway that couples both organic and inorganic C cycles in a wider range of terrestrial ecosystems than previously thought.



**Author contribution**

CR: data curation, formal analysis, funding acquisition, investigation, visualization, writing – original draft. EPV:
conceptualization, methodology, supervision. SB: conceptualization, methodology, writing – original draft. GC: data
curation, formal analysis, software, visualization, writing – original draft. AR: funding acquisition. MA: investigation. SD:
investigation. TJ: formal analysis, software. NR: funding acquisition, investigation. PV: methodology. MCR:
conceptualization, data curation, formal analysis, funding acquisition, methodology, project administration, supervision,
visualization, writing – original draft. All authors: writing – review & editing.

**Competing Interests**

The authors declare that they have no conflict of interest.

**Acknowledgments**

This research was primarily funded by Sadhana Forest, for which we are immensely grateful. We would also like to express
our sincere appreciation to the Agassiz Foundation (University of Lausanne, Switzerland), which provided financial support
for the fieldwork. Additionally, we would like to extend our gratitude to Dr. Caroline De Meyer, Laetitia Monbaron,
Brandon Quinn, Jan Isler, Saskia Petit, Dr. Brahimsamba Bomou, Dr. Benita Putlitz, Dr. Jorge Spangenberg, and Prof.
Torsten Vennemann for their invaluable assistance with the project. Thanks to Mr S. R. Raja, Under Secretary (PP-II), who
authorised the sample export. Finally, we would like to express our gratitude to all the volunteers at Sadhana Forest who
assisted with the fieldwork.

**Financial support**

This work was supported by Agassiz Foundation (University of Lausanne, Switzerland) and Sadhana Forest (India).

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
