# Peer review of "Novel oxalate-carbonate pathways identified in the tropical dry evergreen forest of Tamil Nadu, India"

_EGUsphere, 2025_

## Author Comment (AC1)

This study aimed to assess botanical, microbiological, biogeochemical evidence for the oxalate carbonate pathway in the Dry Evergreen Tropical Forest of Tamil Nadu, India. This pathway is interesting as it can potentially convert organic carbon into inorganic calcite which can be a highly stable form of carbon, depending on the environment.

I thoroughly enjoyed reading this paper, it is a very interesting and important piece of research. Quantifying organic-inorganic carbon conversions is necessary to establish if processes like the OCP contribute substantially to global carbon cycles. This is particularly important in an environment of ecosystem disturbance where these cycles could be disturbed or stopped.

My main criticism of the work is already highlighted by the authors in their discussion and that is the shallow sampling depth limits interpretation. Thus, the possibility of deep pedogenic or geogenic carbonates could not be ruled out which makes the interpretation difficult. Despite this I think the paper is worthy of publication and makes an important contribution.

Dear Prof. Dr Catherine Clarke,

We sincerely appreciate your review of our article and constructive feedback. Your suggestions were highly valuable and have clearly strengthened the manuscript. In particular, we really appreciate your insights and comments on the importance of termites in the OCP. Please find our detailed responses to your individual comments below.

A few questions:

1) Did the parent materials initially contain carbonates? I could not find this information in the supporting Information, but given their origin I guess it is possible? I see Calcisols are mentioned in the underlying Manaveli clay (Lines 155/6). I think this is quite important to mention as it affects the Ca source and calcite cycling.

This was a great point, and it was removed to the supplementary methods during the editing process. The Manaveli clay does contain some poorly preserved mollusc shells and rhizoliths. The Manaveli clay and its associated Calcisols are only exposed at very low elevations across the landscape, away from our sampling areas. The Calcisols were formed from the erosion, exposure, and then subsequent weathering of these poorly preserved molluscs and rhizoliths; the weathering products of which, then precipitated as secondary carbonates in the soil profile. Meanwhile the overlying Cuddalore Sandstone is acidic in nature, doesn't contain any carbonates, and is separated from the Manaveli Clay by a layer of Petroplinthite. The Cuddalore sandstone is an acidic iron-rich sediment formed from the charnokite in the Eastern Ghats and deposited in an estuarian environment of mixed velocity. Our sampling sites were in the Alluvium or Cuddalore sandstone (S.Fig. 10-12) and thus the calcium available to our sampled trees predominantly originates from aeolian deposition and weathering of non-calcareous parent material.

To address any potential confusion, we have added this to the main text:

"While the Manaveli clay contained some poorly preserved molluscs and calcareous rhizoliths, the overlying Cuddalore sandstone is a non-calcareous, acidic, iron-rich sandstone (Fig. 1D). We can thus assume that Ca cycled by the trees was sourced from non-calcareous sources."

> 2) Related to this, I was also wondering about the influence of termites. I assume they exist in this environment? How did the soil enter the trunk? In other environments it is often through termite activity and if so, is there the possibility that calcite-rich subsoils were brought up into the trunks as castings? S. Fig1-C looks a bit like it could be a termite casting? More information on the trunk soil and its potential source is very important for the study, as this is one of the main pieces of evidence that supports the hypothesis.

Another great point, thank you. Termites (particularly *Trinervitermes* spp.) are abundant and active in this environment, and they certainly play a key role in the OCP (Clarke et al., 2023; Nel et al., 2025b, a). We had a similar hypothesis based on the literature, suggestions from our colleague (at the time) John Van Thuyne, and specifically HCl tested the local termite nests for $CaCO_3$. Unfortunately, we could not find any evidence of $CaCO_3$ precipitation associated with the termite species, their nests, or the associated trees that we tested (in various locations at each site). If termite activity were the main driver of the $CaCO_3$ precipitation, we would expect comparable carbonate concentrations in other termite nests and the trees associated with them throughout the site, and this was not the case.

While we cannot rule out the transport or decomposition of the trunk environments by termites, there was no active termite nests associated with these trees. Furthermore, it is unlikely that termites brought up $CaCO_3$ rich soil from the Manaveli clay and precipitated it in castings in the trunk of trees in soils on the Cuddalore sandstone; partially as the two layers are separated by petroplinthite and as we found a similar precipitation in trunk environments well away from the Manaveli clay on the alluvium.

We however thank you for your comment and raising this point, and we have added a text about the importance of termites to the OCP and their presence at our site in the introduction and discussion.

"Instead of accumulating in soils, when CaOx crystals are released during litter decomposition (Hervé et al., 2018), termite activity (Clarke et al. 2023, Nel et al. 2025a, 2025b), or belowground via root turnover or exudation (Cailleau et al., 2014; Rowley et al., 2017), they can readily be metabolised by oxalotrophic microorganisms."

"While we cannot rule out the possibility that termites played a role in the accumulation of CaOx-rich organic matter and precipitation of $CaCO_3$ in the trunks (Clarke et al. 2023, Nel et al. 2025a, 2025b), tests on local termite nests and the trees associated with them did not identify the presence of termite-associated $CaCO_3$. Driven by an active OCP, Pons et al. (2018) reported

a similar $pH_{H2O}$ increase of +1.5 pH units at the foot of M. excelsa, compared to control soils located 30 m away."

There are a few more specific comments below

Lines 90-95: it feels like locations could be listed more eloquently.

We have adjusted the list here:

"Active OCPs have been reported across continents (S.Table 1): in South (Cailleau et al., 2024), Central (Álvarez-Rivera et al., 2021; Rowley et al., 2017) and North America (Garvie, 2006, 2003), in Africa (Aragno and Verrecchia, 2012; Cailleau et al., 2005; Hervé et al., 2021; Pons et al., 2018), and Asia (Verrecchia, 1990; Paper from China Termites; Hervé et al., 2018)."

Lines 100 – 105: I like hypothesis testing, but I would leave aspects like underexplored and undetected out of the central hypothesis, those are hard to test. The hypothesis should just read ...... the OCP is a substantial C sink in India's TDEF ecosystems. This can be tested. The underexplored and undetected can be added elsewhere to highlight the research gap.

Great suggestion, we have simplified the statement accordingly:

"We hypothesise that the OCP represents a substantial C sink in India's TDEF ecosystems.

To explore OCP ecosystems within restored- and primary-TDEF in Tamil Nadu, India, an initial field survey was conducted."

Section 2.2.1: I would appreciate the explanation of "Dry" in the name of the TDEF, the rainfall is very high, so is dry used to highlight the ustic climate with a dry winter? When I read the title, I was thinking semi-arid until I saw the rainfall data.

Yes, exactly, this is a technical term for the forests through the Coromandel coast. While the environment has, annually, two separate monsoon periods and receives significant precipitation, it is also characterised by a marked dry season. In the vegetation classification, the term "dry" in TDEF indeed refers to the dry season during winter, whereas Tropical Moist Evergreen Forests are found for example on the Indian West Coast.

I am a little confused by Figure 1. If I look at site D in Fig 1B then it looks like it is on the sandstone, but in 1C it is shown as alluvium?

Thank you for pointing out this potential source of confusion. Site C is shown in Fig. 1C (Alluvium) and Site D is shown in Fig 1D (Cuddalore). We tried to add arrows to the map to indicate this more clearly (see below), however it unfortunately clutters the map and we have instead tried to make this clearer in the caption and kept the figure as is.

[Figure]

Adjusted figure caption below:

"(C) The Puthupet Sacred Grove sampling site (Site C in Fig. 1B). (D) The Sadhana Forest sampling site (site D in Fig. 1B)."

Lines 150 – 155: Could you mention if the parent materials contained calcite in them? Also maybe expand on the clay soils that can contain calcisols?

We have now added to this to the text as per above.

Section 2.2.2: I think there needs to be more information on the control soils? Was it bare soil/grassland or were there other tree species. Was the surrounding vegetation checked for oxalates? Also, some pictures in SI of the trees and controls would be great.

Thanks this is a great suggestion and we have added a supplementary table with pictures of the studied trees in the SI. We have also clarified in the text that control soils were collected in open areas (not always grassland):

"Hence, with explicit permission, surficial samples were instead collected to a depth of 10 cm near the trees (adjacent) and a control distance of 20 m away from the tree in an open area (control)."

We did screen 25 tree species (S.Table 4) for oxalate content in leaves and carbonate deposits (Ilarslan et al., 2001). It indicated that most tree species in our specific sample area contain only low oxalate concentrations compared to our select species, and therefore their potential contribution to carbonate production was expected to be minimal. However, more detailed analyses of numerous species in the TDEF would be required to quantify the production of oxalate in these forests.

Lines 160-165: Some more information on the trunk soil is required, is it the same colour and texture as the adjacent topsoil? How did it get there? Termites? They often bring deep

subsoil material up to build protective castings in trees. If so, this might be an alternative explanation of how the carbonates got there.

The soil material here has a much higher C content and is much darker, formed from the decomposition of the interior of the still alive tree. As highlighted above, there were no active termite nests associated with the trees nor any evidence of $CaCO_3$ precipitation associated with the local termite species. Furthermore, two of three of these trunks were located on the Alluvium deposits, much further away from the influence of the Manaveli Clay.

To clarify, we added a phrase about it:

"These soils are enriched in organic matter, as they are derived from the decomposition of the surrounding tree's wood."

Lines 170-175: The soils were milled, but it is unclear what these soils were used for?

We have changed this sentence to make it clearer:

"For analyses requiring powdered material, a subsample of each sample was ground accordingly."

Lines 120-180: It is mentioned that plant material was milled prior to SEM analysis. Were the samples rinsed prior to milling to remove dust particles. I am not sure if windblown marine dust aerosols could add CaCO3 to plant parts?

Thank you for spotting this important typo. The plant samples prepared for SEM were not milled, as our objective was to observe CaOx crystals and oxalotrophs in situ within the plant tissues. Milling was only applied to subsamples that were used for the quantification of CaOx with enzymatic kits and to increase contact with the extractant. The SEM observations are not affected by potential external $CaCO_3$ contamination from dust particles.

Lines 195 – 200: I see cobalt hexamine was used for exchangeable cation extraction, with the reason that it is less aggressive towards CaCO3. It is also worth mentioning that Ca oxalate is also susceptible to aggressive extraction agents.

Thanks for this important point. We added a sentence to enhance it:

"This method was selected because it displaces exchangeable cations while minimising dissolution of phases such as $CaCO_3$ or CaOx".

Line 214: was should be were

Corrected, thank you.

S. Table 7: I think the sum of basic cations have been calculated incorrectly for this table?

Thank you so much this was a copying error from our Master table and we appreciate your thorough review, even of our supplementary materials. We have removed this column from our table and gone back and checked all our other supplementary data.

Lines 380-385: When you talk about the CEC are you referring to sum of total cations? This would be best described as effective cation exchange capacity (ECEC). CEC is measured in a buffered salt usually at pH 7? It is remarked that the CEC is highest in the trunk, and this was driven by higher Ca and Mg concentrations. I think the choice of words here is incorrect. CEC is not driven by cations, rather a soil can accommodate more cations because it has more exchange sites. Was there a textural difference between the soils in the trunk and the adjacent soils, more clay maybe? Looking at the substantially higher SOC in the trunk, I think the higher ECEC is probably related to the exchange sites on the SOC? Is the SOC higher in the trunk because it contains decaying plant material?

Thank you, you are completely correct that CEC is the sum of exchange sites and we should have been more precise in our terminology, we will use effective cation exchange capacity throughout. Here we measure the sum of exchangeable cations, but not exchangeable acidity. There was no real difference in the texture of soils, but this is instead, as you highlight, caused by the increased SOC content; and thus, the increased proportion of negatively charged organic functional groups, leading to the higher retention of $Ca^{2+}$ and $Mg^{2+}$.

My final point on the cation data is: Given how large the difference is between the pH in KCl and DI, I am surprised that the exchangeable Al is so low. I am not familiar with Co hexamine, but usually exchangeable Al+ $H^+$ is measured in a 1M KCl extract. I mention this as Al does feature in your discussion later and the reserve acidity associated with exchangeable Al may be the reason for the similar pH of the control and adjacent soils.

We were also slightly surprised by this, but checked our measurements, going back to raw data and recalculating everything prior to preparing the manuscript. We used a similar extraction in Rowley et al. (2020) (https://doi.org/10.1016/j.geoderma.2019.114065) and you'll see that CoHex extraction does also extract Al when it is present in acidic soils (here acidic Eutric Cambisols; range of $CaCO_3$ free soils 0.8±SE0.1 cmolc. kg$^{-1}$ range: 0.1-1.5 cmolc. kg$^{-1}$). You're right that the 1 M KCl extract would have been good to extract specifically exchangeable acidity here but would have concomitantly prevented the measurement of $K_{Exch}$.

Considering your comments and out of interest, Dr. Rowley went back to this old dataset and compared the Al extracted by CoHex and the **2M** KCl extract, which was concomitantly measured during the investigation of Ca extracts (Fig. 4). In these alpine acidic soils, the extraction efficiency of 2M KCl was slightly higher than CoHex (+12 ± SE 5 % Al ranging from –5 to +30 %) and we can therefore assume that the CoHex extraction would have roughly the same extraction efficiency as **1 M** KCl in these soils, although this would require confirmation. In future, it would be a good idea to investigate the pyrophosphate, oxalate, and

citrate bicarbonate dithionite extracted Al in these soils too, to further probe shifts in Al dynamics.

Lines 544 – 545: It is noted that the current study area has lower rainfall than the site used by Herve et al (2018), but the rainfall described in the methods would suggest its higher (1225 vs 1100 mm/y)?

Thank you for spotting this. We have corrected it as follows:

"The present study lends further support to this idea, as while the area is subject to two separate seasons, with dry (inter-monsoons) and wet (monsoons) periods, conditions were slightly moister in Tamil Nadu than in Madhya Pradesh (Duraisamy Rajasekaran et al., 2024; Hervé et al., 2018; Saikranthi et al., 2024)."

Finally just a comment: It is really interesting that the pH of the adjacent soils and the control soils were no different, you would think over time, there would be some changes even to the topsoils. Maybe another metric to include in these high rainfall regions would be exchangeable acidity or some estimation of reserve acidity. This would then show if any sort of pH buffering was occurring in the adjacent soils.

Yes, exactly, either exchangeable acidity or the Al extracts mentioned above. It would be great to also investigate changes in Al speciation with more advanced methods too. We added a sentence about this at the end of 4.3. as a conclusion:

"In high-rainfall regions, pH alone may not capture subtle buffering effects in surface soils. Future studies could include measures of exchangeable acidity, which could provide insight into pH buffering dynamics in adjacent versus control soils."

**References**

Clarke, C. E., Francis, M. L., Sakala, B. J., Hattingh, M., and Miller, J. A.: Enhanced carbon storage in semi-arid soils through termite activity, CATENA, 232, 107373, https://doi.org/10.1016/j.catena.2023.107373, 2023.

Nel, T., Clarke, C. E., Francis, M. L., Babenko, D., Botha, A., Breecker, D. O., Cowan, D. A., Gallagher, T., Lebre, P., McAuliffe, J. R., Reinhardt, A. N., and Trindade, M.: Carbon dynamics in termite mounds: The effect of land use on microbial oxalotrophy, CATENA, 254, 108947, https://doi.org/10.1016/j.catena.2025.108947, 2025a.

Nel, T., Clarke, C. E., Francis, M. L., Babenko, D., Breecker, D., Cowan, D. A., Gallagher, T., McAuliffe, J. R., and Trindade, M.: Oxalate content of vegetation and termite frass in western South Africa, Ecosphere, 16, e70265, https://doi.org/10.1002/ecs2.70265, 2025b.

Rowley, M. C., Grand, S., Adatte, T., and Verrecchia, E. P.: A cascading influence of calcium carbonate on the biogeochemistry and pedogenic trajectories of subalpine soils, Switzerland, Geoderma, 361, 1–12, https://doi.org/10.1016/j.geoderma.2019.114065, 2020.

---

## Author Comment (AC2)

The presented work is excellent, with a complete state-of-the-art on the Oxalate-carbonate pathway presented at full, a clear local Indian/monsoon context, and a transparent experimental design. The results are essential for supporting understanding of our planet's natural carbon removal mechanisms, via plant-microbial-soil interactions, that can be supported in OCP-enabling forest ecosystems, as long as Ca sources are additional, and plant oxalate inputs sufficient for alkalinity.

I find the overall quality of the work to be stellar, with the exception of low replication, though the authors have dealt this with transparent and appropriate statistical methods. In my sense, this work should be considered as a primer, for larger extensive field studies, supported by local Indian scientists and beyond. It showcases the potential of the OCP. Other than that I have no other criticisms. What I find exciting is the potential quantities stored as calcium carbonate in bark, with perhaps another avenue to explore , using bark content as another stable inorganic carbon stock (easily measurable, and verified), in additional soil carbonates, and flushed bicarbonate. Perhaps the authors could dedicate a few sentences in their discussion to showcase a comparison between inorganic carbon stock (in bark), and potential bicarbonate flushing underground. These are essential for project developers, and MRV requirements.

None technical corrections noted.

Dear anonymous reviewer 2,

We sincerely thank you for the very positive and encouraging feedback on our work. We hope that this work is only, but the first step, in a broader collaboration with other local colleagues to further investigate the OCP in India. We agree that bark represents a relevant stable inorganic carbon stock, in addition to bicarbonate flushing, but that these would need to be quantified in more detail. We have therefore added a cautious point in the discussion to underline its importance for future research, at the end of 4.4 paragraph:

"In addition, $CaCO_3$ precipitation, inorganic carbon stocks, and bicarbonate export both above and belowground deserves further investigation in any monitoring, reporting, and verification of OCP-systems."

We thank you for your review of our article.